# SAR-SLAM: Self-Attentive Rendering-based SLAM with Neural Point Cloud Encoding

## ABSTRACT

Neural implicit representations have recently revolutionized simultaneous localization and mapping (SLAM), giving rise to a groundbreaking paradigm known as NeRF-based SLAM. However, existing methods often fall short in accurately estimating poses and reconstructing scenes. This limitation largely stems from their reliance on volume rendering techniques, which oversimplify the modeling process. In this paper, we introduce a novel neural implicit SLAM system named SAR-SLAM to address these shortcomings. Our approach reconstructs Neural Radiance Fields (NeRFs) using a self-attentive architecture and represents scenes through neural point cloud encoding. Unlike previous NeRF-based SLAM methods, which depend on traditional volume rendering equations for scene representation and view synthesis, our method employs a self-attentive rendering framework with the Transformer architecture during mapping and tracking stages. To enable incremental mapping, we anchor scene features within a neural point cloud, striking a balance between estimation accuracy and computational cost. Experimental results on three challenging datasets show the superior performance and robustness of our SAR-SLAM compared to recent NeRF-based SLAM systems. The code will be released.

## CCS CONCEPTS

• **Computing methodologies** → **Artificial intelligence**; **Vision for robotics**.

## KEYWORDS

Simultaneous Localization and Mapping, Self-Attentive Rendering, Neural Point Cloud

**ACM Reference Format:**
Anonymous Author(s). 2018. SAR-SLAM: Self-Attentive Rendering-based SLAM with Neural Point Cloud Encoding. In *Proceedings of Make sure to enter the correct conference title from your rights confirmation emai (Conference acronym 'XX)*. ACM, New York, NY, USA, 10 pages. https://doi.org/XXXXXXX.XXXXXXX

## 1 INTRODUCTION

Simultaneous Localization and Mapping (SLAM) [4] stands as a fundamental challenge in computer vision, finding wide-ranging applications in autonomous driving [3], robot navigation [5], augmented reality (AR), virtual reality (VR), collision detection [7], and

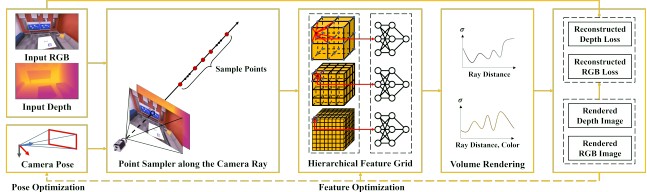

(a) With traditional volume rendering (NICE-SLAM)

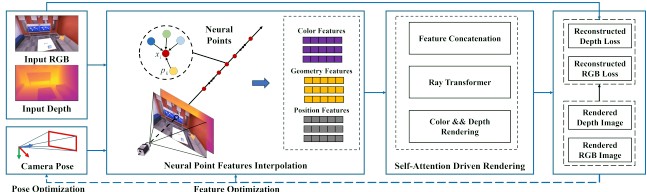

(b) With generalizable NeRF transformer rendering (Ours)

**Figure 1: The comparison of the NeRF-based SLAM with different image rendering methods.**

scene perception [27]. With the escalating demand for high-fidelity 3D scene reconstruction, there's a critical shift towards representing target scenes more accurately, favoring continuous surfaces over discrete point clouds. Despite the remarkable strides made in 3D scene reconstruction technology in recent years, achieving real-time representation of high-quality scenes without compromising accuracy and spatial resolution remains a persistent challenge.

The advent of low-cost visual sensors has led to the emergence of numerous real-time dense visual SLAM systems, gaining substantial attention across academia and industry. Traditional methods in dense visual SLAM employ representations such as point clouds, surfels, voxel grids, voxel hashing, or octrees to achieve real-time, dense, and large-scale scene reconstruction [30, 47]. However, these methods often struggle to provide accurate geometric estimations for unobserved areas. In contrast, learning-based SLAM methods exhibit predictive capabilities as they are typically trained on task-specific datasets [1, 16, 38, 43, 48]. Yet, many of these methods rely on ground truth depth or 3D mesh for training, posing challenges in generalization to unseen scenes during testing. Furthermore, deep learning-based scene representation encounters challenges with local scene updating and fixed network capacity during runtime, constraining their capability for achieving high-fidelity reconstruction in large scenes.

With the introduction of Neural Radiance Fields (NeRF) [20] and its diverse applications in inverse rendering [32], controllable editing [23], digital human body generation [40], multi-modality [37], image and video processing [15], medical imaging [31], and various other fields, researchers have begun integrating NeRF into SLAM

research. The implicit representation produced by the continuous radiance field enables high-quality rendering of both visible and occluded regions, facilitating the extraction of underlying shapes at any resolution. Compared to conventional methods, implicit neural scene representations demonstrate superior noise and outlier suppression capabilities in mapping, enhanced hole-filling and inpainting capacities for occluded scenes, and more robust data compression abilities. Despite showcasing significant performance advancements, these approaches often involve decoding the feature volume into a radiance field and rely on classic volume rendering formulas for view synthesis. It's important to note that the volume rendering equation utilized in NeRF oversimplifies optical modeling concerning solid surfaces [45], reflectivity [6, 8], and inter-surface scattering effects. Consequently, within NeRF, the brightness field associated with volume rendering does not present a universal imaging model, thereby limiting the generalizability of NeRF-based SLAM algorithms when encountering new scenes.

Drawing inspiration from IBRNet [39], we present SAR-SLAM, a NeRF Transformer-based SLAM system with neural point cloud encoding designed for improving generalizability. As illustrated in Fig. 1, our approach hinges on several key ideas. 1) Instead of relying on classical volume rendering equations for view synthesis, we employ a ray transformer that utilizes self-attention mechanisms to compose coordinate-wise point features along traced rays. This departure allows for inducing view synthesis in a more effective manner. 2) To balance memory usage and accuracy, we leverage scene-adaptive neural point clouds for scene representation [42]. Each neural point encapsulates both geometric and color features of the local map. During rendering, we employ scene-adaptive ray-based rendering facilitated by learned attention mechanisms, enabling the translation of these features into scene geometry and color estimates. 3) By utilizing the depth and color images rendered from the ray transformer outputs, we optimize scene representations and camera poses by minimizing re-rendering losses throughout the tracking and mapping processes. Our method is extensively evaluated across synthetic and real-world datasets, showcasing advancements in tracking, mapping, and rendering. In summary, the contributions of this paper are three-fold:

(1) We propose SAR-SLAM, a novel neural implicit SLAM system designed to operate in real-time and demonstrate robustness across diverse challenging scenarios of various scales.

(2) SAR-SLAM's core innovation lies in the implicit attention-based ray rendering idea. Differing from the traditional volume rendering equation, this approach reconstructs NeRF without requiring an explicit rendering formula, thereby enhancing strength, scalability, and versatility in graphical rendering.

(3) Comprehensive evaluations performed across various datasets validate the superiority of our approach in terms of tracking, mapping, and rendering capabilities.

## 2 RELATED WORK

**Dense Visual SLAM.** In recent years, visual SLAM has seen significant activity in both academia and industry. Unlike sparse visual SLAM algorithms that utilize sparse point clouds for scene map representation and camera pose estimation, dense visual SLAM algorithms leverage dense point clouds, meshes, or accelerated grids

to reconstruct detailed scene maps. Generally, map representations fall into two categories: view-centered and world-centered. The former often represents 3D geometry as depth maps of keyframes, as seen in approaches like DTAM [22], ORB-SLAM [21], and subsequent works. DTAM's straightforward pipeline is widely used in SLAM systems employing deep learning for depth and pose estimation. Similarly, DeepFactors [9] simplify optimization by employing a collection of basic depth maps. Other methods, such as CodeSLAM [2], SceneCode [49], and NodeSLAM [35], optimize latent representations decoded into keyframes or object depth maps. DROID-SLAM [36] utilizes regression optical flow for geometric refinement, while TANDEM combines multi-view stereo with DSO for real-time dense SLAM. DeepSLAM [17] utilizes autoencoder networks and recurrent convolutional neural networks to predict scene depth and 6DoF pose, respectively. Alternatively, another approach employs a world-centered map representation that anchors 3D geometry in unified world coordinates, representing scene maps as surfels [30, 41] or occupancies/TSDF values in voxel grids [11, 24].

**NeRF-based Visual SLAM.** Thanks to neural implicit representations facilitating object-level reconstruction [46], scene completion [19, 25], new view synthesis [26], and various other aspects, several NeRF-based Visual SLAM methods aim to jointly optimize the neural radiance field and camera pose. BARF [18] utilizes a neural rendering network for implementing Bundle Adjustment (BA), a crucial process in traditional SLAM systems, executing iterative optimizations for both model and camera pose. iMAP [34] utilizes the neural rendering model to establish two threads: tracking and mapping. The tracking thread leverages current model parameters to reason and optimize the camera pose of the current frame, while the mapping thread refines the network model parameters and camera pose after integrating new keyframes. Building upon iMAP, NICE-SLAM [51] employs feature grids to encode the scene into multi-dimensional vectors and utilizes an MLP to decode the implicit representation into occupancy and color. Further advancements, such as NICER-SLAM [50], have proposed a dense RGB-SLAM system that concurrently optimizes camera poses and multi-level neural implicit representations, enabling high-precision positioning and the synthesis of new views with exceptional fidelity. Similarly, in literature [29], the exclusive use of RGB images as input introduces a photometric consistency transformation error based on multi-view geometric constraints, resulting in enhanced constraints for camera pose estimation and the geometric structure of the scene.

Unlike the NeRF-based Visual SLAM approaches mentioned earlier that rely on classic volume rendering for view synthesis subsequent to feature encoding or aggregation, our proposal introduces a self-attentive ray transformer to model this process. This methodology constructs a more versatile imaging model for the SLAM system, capable of synthesizing higher-quality images within restricted viewing angles. Consequently, it enhances the accuracy of localization and mapping of NeRF-based SLAM algorithms. Diverging from methods like NICE-SLAM that utilize multi-scale feature grids to represent the scene, our approach involves neural point cloud encoding [28, 42]. The neural points offer a user-friendly representation, facilitating faster neighborhood search and achieving a balanced trade-off between accuracy and efficiency.

**Figure 2: System Overview. Our proposed SAR-SLAM takes RGB-D image sequences as input and concurrently generates camera poses alongside a learned scene representation using neural point clouds. Upon receiving each new RGB-D image, we incorporate a set of neural points using a specific sampling approach. Subsequently, we employ a versatile ray transformer to render depth and color maps. The sampled point features encompass three components: color and geometry features derived from the neural point cloud map interpolation, and positional features from learnable Gaussian position encoding. Following the generation of rendered images (depth and RGB), we estimate camera poses and refine the scene representation through an iterative optimization process that minimizes re-rendering losses. This iterative process involves the optimization of neural point features and the ray transformer network during mapping. In an alternating manner, we optimize camera poses during tracking while keeping the map and network fixed.**

## 3 METHOD

The overview of our proposed method is depicted in Fig. 2. We use neural point clouds to represent the scene, which is incrementally added during the exploration process (Sec. 3.1). In contrast to previous NeRF-based SLAM approaches that optimize scene representation using fixed volume rendering equations, SAR-SLAM leverages learnable self-attentive ray transformers for view synthesis (Sec. 3.2). Throughout mapping and tracking, we minimize re-rendering losses related to depth and color. This simultaneous optimization enhances both camera poses and scene representation (Sec. 3.3 ).

### 3.1 Scene Representation with Neural Point Clouds

Differing from the majority of NeRF-based SLAM methods that use a hierarchical grid for scene representation, we integrate geometry and color features into the neural point cloud using scene-adaptive point density, as proposed in [42]. We define a set of neural point clouds with $N$ points as follows:

$$P = \{(p_i, f_i^g, f_i^c) | i = 1, \ldots, N\}, \tag{1}$$

where $p_i \in \mathbb{R}^3$ is the location of the anchored point, $f_i^g \in \mathbb{R}^{32}$ and $f_i^c \in \mathbb{R}^{32}$ are the geometric and color feature descriptors respectively. During the mapping process, we conduct uniform sampling and large gradient pixel sampling on RGB images. If the depth value corresponding to the sampled pixel is valid, the 2D pixel is projected onto the 3D space. Subsequently, neighboring neural points within the specified search radius are identified within the neural point cloud. The density of the neural point cloud is intricately linked to the chosen search radius, which is governed by the color gradient-based dynamic resolution strategy outlined in [28].

In scenarios where no neural point is found in the vicinity, we sample three points aligned with the depth value $D$ along the camera ray: $(1 - \rho)D$, $D$, and $(1 + \rho)D$, where $\rho \in (0, 1)$ serves as a

hyperparameter, accounting for the anticipated depth noise. The feature vectors of these newly added neural points are initialized using Kaiming initialization [12]. As more frames are processed, the neural point cloud expands progressively to encompass the exploration of the scene. It tends to converge to a finite set of points once all parts of the scene have been accessed. Unlike many grid-based scene representations, this approach doesn't necessitate the specification of scene boundaries before reconstruction.

### 3.2 Self-Attentive Rendering

Current NeRF-based SLAM methods employ a fixed volumetric rendering pipeline to synthesize novel view images for poses and scene representation optimization. This pipeline combines the color and density of points along each ray cast from the image plane to produce the final pixel color. Given camera intrinsic parameters and the estimated camera pose, we can obtain the camera ray $\mathbf{r} = (\mathbf{o}, \mathbf{d})$, where $\mathbf{o}$ is the origin and $\mathbf{d}$ is the ray direction. We sample a set of points $x_i$ along this ray as:

$$x_i = \mathbf{o} + z_i \mathbf{d}, i \in \{1, \ldots, N_s\}, \tag{2}$$

where $z_i \in \mathbb{R}$ is the point depth. For pixels with valid depth value $D$, we sample 5 points uniformly between the limited band $(1 - \rho)D$ and $(1 + \rho)D$. With this scene depth prior, we can sample fewer samples along the ray, which achieves a computational speed-up during rendering.

The volume rendering can be conceptualized as a weighted aggregation of all point-wise outputs, where the weights are globally determined by points along traced rays for occlusion modeling. We argue that this aggregation process can be effectively learned by a transformer model, referred to as the ray transformer in this paper. Specifically, the point-wise colors are mapped to token features, while the attention scores correspond to transmittance, representing the blending weights in the rendering process.

The detailed network architecture of the ray transformer is depicted in Fig. 3. Following the point sampling, we combine the

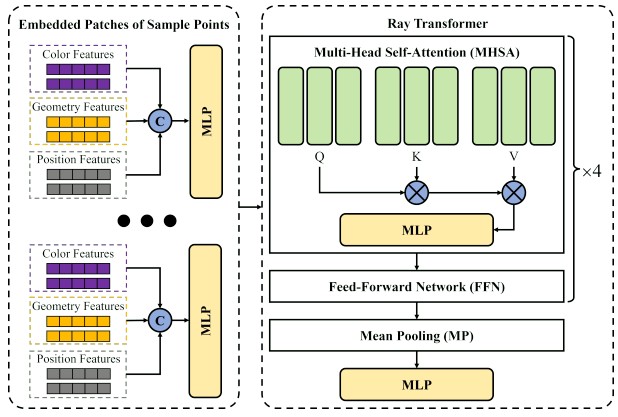

**Figure 3: Detailed network architecture of ray transformer in SAR-SLAM.**

geometry features $F^g(x_i)$, color features $F^c(x_i)$, and positional features $F^p(x_i)$ of the point $x_i$ to form $F_i$ via Feature Concatenation (FC). Subsequently, by inputting the sequence $\{F_1, F_2, F_3, F_4, F_5\}$ into the ray transformer, we conduct mean pooling over all the predicted tokens. Finally, we map the pooled feature vector to RGB through a Multilayer Perceptron (MLP):

$$I(x_i) = MLP \circ MP \circ Transformer(F_1, F_2, F_3, F_4, F_5), \quad (3)$$

where MP represents the mean pooling operation. For each point $x_i$, we find its closest eight neighbor points $\{p_k, k = 1, \ldots, 8\}$ using the corresponding per-pixel query radius, and use inverse squared distance weighting for features trilinear interpolation:

$$F^g(x_i) = \sum_{k=1}^{8} \frac{w_k}{\sum_k w_k} f_k^g, w_k = \frac{1}{\|p_k - x_i\|_2}, \quad (4)$$

$$F^c(x_i) = \sum_{k=1}^{8} \frac{w_k}{\sum_k w_k} f_{k,x_i}^c,$$

$$w_k = \frac{1}{\|p_k - x_i\|_2}, f_{k,x_i}^c = \mathcal{F}_\theta(f_k^c, p_k - x_i), \quad (5)$$

where $f_k^g$ and $f_k^c$ are geometry features and color features of neural point $p_k$ respectively, $\mathcal{F}_\theta$ is a one-layer MLP inspired by [42]. For pixels without valid depth, we sample 30 points along the ray within a depth interval. As an additional output of the ray transformer, the weights $\alpha_i$ of each predicted token can be used for depth $\hat{D}$ rendering and variance $\hat{S}_D$ computation along the ray as:

$$\hat{D} = \sum_{i=1}^{N_s} \alpha_i z_i, \hat{S}_D = \sum_{i=1}^{5} \alpha_i (\hat{D} - z_i)^2, \quad (6)$$

where $N_s = 5$ for sampled pixels with valid depth and $N_s = 30$ for invalid situation. This framework demonstrates superiority in view synthesis compared to other approaches lacking a self-attentive scheme (*e.g.*, pure MLP. See Table 9 in Sec. 4.4).

## 3.3 Mapping and Tracking

In this section, we delve into the optimization specifics concerning the color features and geometry features linked to the neural point cloud. Additionally, we cover position features encoded through

learnable Gaussian positional encoding, along with insights into the ray transformer network.

**Implicit Mapping.** In the mapping process, we conduct uniform sampling of $M$ pixels from the present RGB-D frame and the chosen keyframes. The re-rendering loss $L_{map}$, comprises geometric loss $L_g$ and photometric loss $L_p$. Both losses are formulated as $L_1$ loss function:

$$L_{map} = \frac{1}{M} \sum_{m=1}^{M} |D_m - \hat{D}_m|_1 + \lambda_{map} |I_m - \hat{I}_m|_1, \quad (7)$$

where $\hat{D}_m$ and $\hat{I}_m$ are the rendered depth and the color value for given ground truth $D_m$ and $I_m$, $\lambda_{map}$ is the loss weighting factor for mapping. This loss function optimizes the geometric and color features, as well as the parameters of the ray transformer and interpolation decoder simultaneously. To ensure effective initialization of the mapping optimization, the iteration count for the first frame is set considerably higher than subsequent frames. Instead of directly assigning iteration parameters, we utilize the difference in Peak Signal-to-Noise Ratio (PSNR) between the rendered image and the ground truth as a criterion. This adaptive thresholding allows us to dynamically adjust iteration numbers for different scenes, resembling the initialization process in classical SLAM systems.

**Camera Tracking.** In a parallel process to mapping, we execute frame tracking to optimize the camera pose $\{R, t\}$ for the current frame. We sample $M_t$ pixels from the current frame and employ a modified re-rendering loss, distinct from Eq. (7), as follows:

$$L_{track} = \frac{1}{M_t} \sum_{m=1}^{M_t} \frac{|D_m - \hat{D}_m|_1}{\sqrt{\hat{S}_D}} + \lambda_{track} |I_m - \hat{I}_m|_1, \quad (8)$$

where $\hat{S}_D$ is the standard deviation of the depth prediction, $\lambda_{track}$ denotes the loss weighting factor for tracking. For simplicity, we initialize the new pose based on a constant speed assumption widely adopted in existing methods [34, 51]. This assumption involves transforming the last known pose using the relative transformation between the second-to-last pose and the last pose.

**Keyframe Selection.** In line with many NeRF-based SLAM systems, we utilize a keyframe database to regulate the mapping loss. We select a subset of keyframes that demonstrate significant overlap with the viewing frustum of the current frame, integrating pixel samples from these keyframes. This selection method ensures efficient optimization for the neural points within the current view and maintains geometric consistency throughout mapping. To begin, we project all neural points onto the current frame using the optimized camera pose to facilitate effective neural point selection. Additionally, we project the neural points onto every keyframe in the global keyframe list and arrange all keyframes in descending order based on the number of projected neural points. Subsequently, we select the top $K - 1$ keyframes from the sorted list to complement the current frame, resulting in a total of $K$ active frames for local bundle adjustment.

The detailed tracking and mapping process of SAR-SLAM is illustrated in Alg. 1. Following system initialization, the tracking and mapping operate simultaneously, optimizing camera poses and scene representation alternatively.

---

**Algorithm 1:** Tracking and Mapping Process of SAR-SLAM

**Input:** RGB-D image sequence $\{I_j, D_j | j = 1, \dots, N\}$
**Output:** Optimized poses $\{\mathbf{T}_j\}$, neural point cloud $P$

1 Neural point cloud generation; $\Rightarrow$ Eq. (1)
2 Adaptive system initialization; $\Rightarrow$ Sec.3.3
3 **for** *each* $j \in [2, N]$ **do**
4     Camera pose initialization;
5     Sample points with depth prior; $\Rightarrow$ Eq. (2)
6     Features extraction; $\Rightarrow$ Eq. (4), Eq. (5)
7     RGB and depth image rendering; $\Rightarrow$ Eq. (3), Eq. (6)
8     Construct tracking loss $L_{track}$; $\Rightarrow$ Eq. (8)
9     Optimize camera pose $\mathbf{T}_j$;
10     **if** $I_j$ *is keyframe* **then**
11        Keyframe selection; $\Rightarrow$ Sec.3.3
12        Construct mapping loss $L_{map}$; $\Rightarrow$ Eq. (7)
13        Optimize $P$ and ray transformer parameters;
14     **end**
15 **end**

---

## 4 EXPERIMENTS

In Section 4.1, we provide an overview of our experimental setup, encompassing datasets, baseline methods, metrics, and implementation details. Subsequently, in Section 4.2, we conduct qualitative and quantitative comparisons with state-of-the-art NeRF-based SLAM methods across synthetic and real-world datasets. Additionally, we offer a thorough ablation study in Section 4.3.

### 4.1 Experimental Setup

**Datasets.** The synthetic dataset Replica comprises high-quality 3D reconstruction of a variety of indoor scenes. We utilize the publicly available dataset collected by Sucar et al [34]., which provides ground truth trajectories and RGBD sequences. Further, we evaluate the performance of our framework in real-world scenes by using TUM-RGBD [33] and the ScanNet [10] dataset. The poses for TUM-RGBD were captured using an external motion capture system while ScanNet uses poses from BundleFusion [11].

**Baseline Methods.** We primarily compare our proposed method against existing state-of-the-art NeRF-based visual SLAM methods, including NICE-SLAM [51], Vox-Fusion [44], ESLAM [14], Uncle-SLAM [29], and Point-SLAM [28]. We derive most comparison results directly from respective papers. For datasets or sequences not available, we reproduce the results using open-source code with default settings.

**Evaluation Metrics.** For tracking, we follow the conventional SLAM evaluation pipeline by aligning the estimated camera trajectory to the GT and using ATE RMSE to evaluate the accuracy. To evaluate scene reconstruction, we produce the meshes by marching cubes and adopt the F-score (harmonic mean of the Precision and Recall) and depth L1 as the primary metrics with a distance threshold of $1cm$. Furthermore, we provide PSNR, SSIM, and LPIPS for rendering evaluation, calculated based on rendering the full-resolution image along the estimated trajectory every $5th$ frame.

**Implementation Details.** We executed our SLAM system on a desktop PC equipped with a 3.80GHz Intel i9-12900K CPU and

an NVIDIA RTX 3090 Ti GPU. For small-scale synthetic datasets (Replica), we select $K = 5$ keyframes for local bundle adjustment, $M = 5,000$ pixels for mapping, and $M_t = 1,500$ pixels for tracking. Conversely, for large-scale real datasets (ScanNet and TUM-RGBD), we adjusted the parameters to $K = 10$, $M = 10,000$ and $M_t = 5,000$. Rather than uniformly sampling pixels, we employed a scene-adaptive scheme that selectively samples half of the pixels based on the image gradient magnitude. Throughout all experiments, we maintained a photometric loss weighting of $\lambda_{\text{map}} = 0.2$ and $\lambda_{\text{track}} = 0.5$. Training is done with the Adam optimizer and the default hyperparameters $\beta = (0.9, 0.999)$, eps=$1e - 08$, and weight decay=0. The learning rates (LR) were set to 0.002 for tracking on Replica and 0.001 on TUM-RGBD and ScanNet. During the mapping stage, we used LR=0.005 for color and depth optimization.

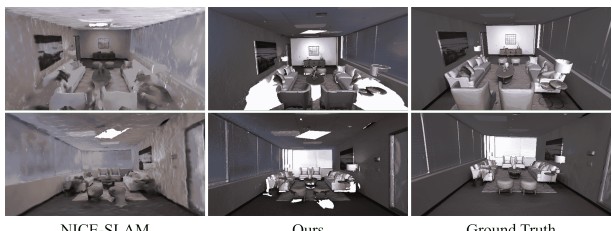

NICE-SLAM        Ours        Ground Truth

**Figure 4: Reconstructed mesh comparison between different NeRF-based SLAM methods.**

| Method | Rm 0 | Rm 1 | Rm 2 | Off 0 | Off 1 | Off 2 | Off 3 | Off 4 | Avg. |
|---|---|---|---|---|---|---|---|---|---|
| ESLAM [14] | 0.71 | 0.70 | 0.52 | 0.57 | 0.55 | 0.58 | 0.72 | 0.63 | 0.63 |
| Vox-Fusion [44] | 1.37 | 4.70 | 1.47 | 8.48 | 2.04 | 2.58 | 1.11 | 2.94 | 3.09 |
| NICE-SLAM [51] | 0.97 | 1.31 | 1.07 | 0.88 | 1.00 | 1.06 | 1.10 | 1.13 | 1.06 |
| Point-SLAM [28] | 0.61 | 0.41 | 0.37 | 0.38 | 0.48 | 0.54 | 0.69 | 0.72 | 0.52 |
| Ours | **0.54** | **0.29** | **0.25** | **0.33** | **0.45** | **0.52** | **0.62** | **0.61** | **0.45** |

**Table 1: Tracking performance on Replica (ATE RMSE ↓[cm]).**

| Method | fr1/desk | fr1/desk2 | fr1/room | fr2/xyz | fr3/office | Avg. |
|---|---|---|---|---|---|---|
| DI-Fusion [13] | 4.40 | - | - | 2.00 | 5.80 | - |
| Vox-Fusion [44] | 3.52 | 6.00 | 19.53 | 1.49 | 26.01 | 11.31 |
| iMAP [34] | 4.90 | - | - | 2.00 | 5.80 | - |
| NICE-SLAM [51] | 4.26 | 4.99 | 34.49 | 31.73 | 3.87 | 15.87 |
| Point-SLAM [28] | 4.34 | 4.54 | 30.92 | 1.31 | 3.48 | 8.92 |
| Uncle-SLAM [29] | 29.04 | 36.57 | - | 5.11 | - | - |
| Ours | **2.79** | **3.08** | **16.86** | **1.16** | **2.89** | **5.36** |

**Table 2: Tracking performance on TUM-RGBD (ATE RMSE ↓[cm]).**

## 4.2 Evaluation of Tracking, Mapping, and Rendering

**Tracking Evaluation.** The tracking results on the Replica dataset are reported in Table 1. SAR-SLAM emerges as the superior performer among all comparison methods across the test sequences. We attribute this success to the introduction of the self-attentive ray rendering and the notably enhanced scene representation afforded by the neural point cloud.

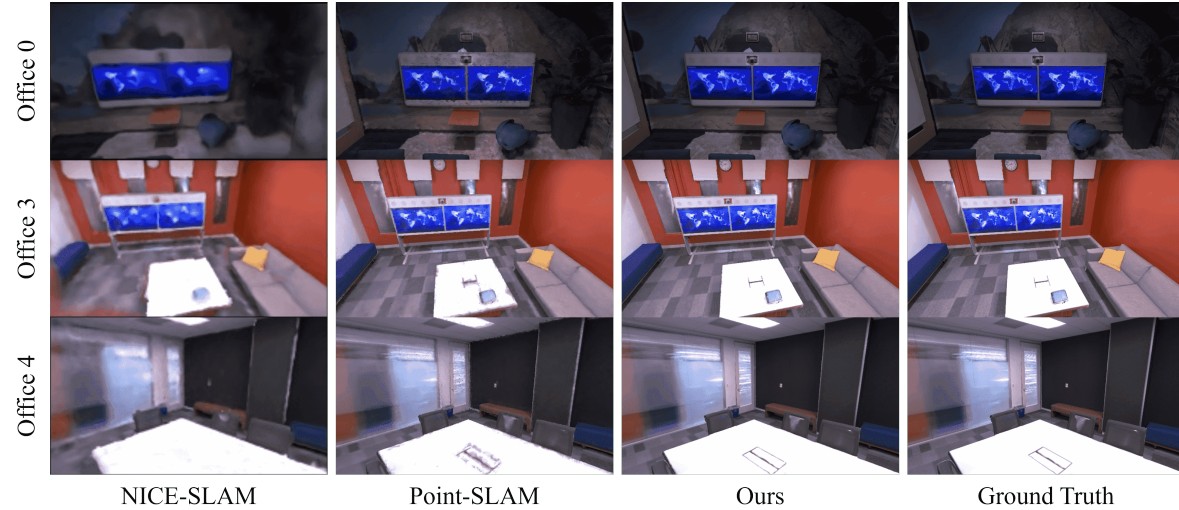

**Figure 5: Novel view synthesis results on the synthetic scenes dataset (Replica).**

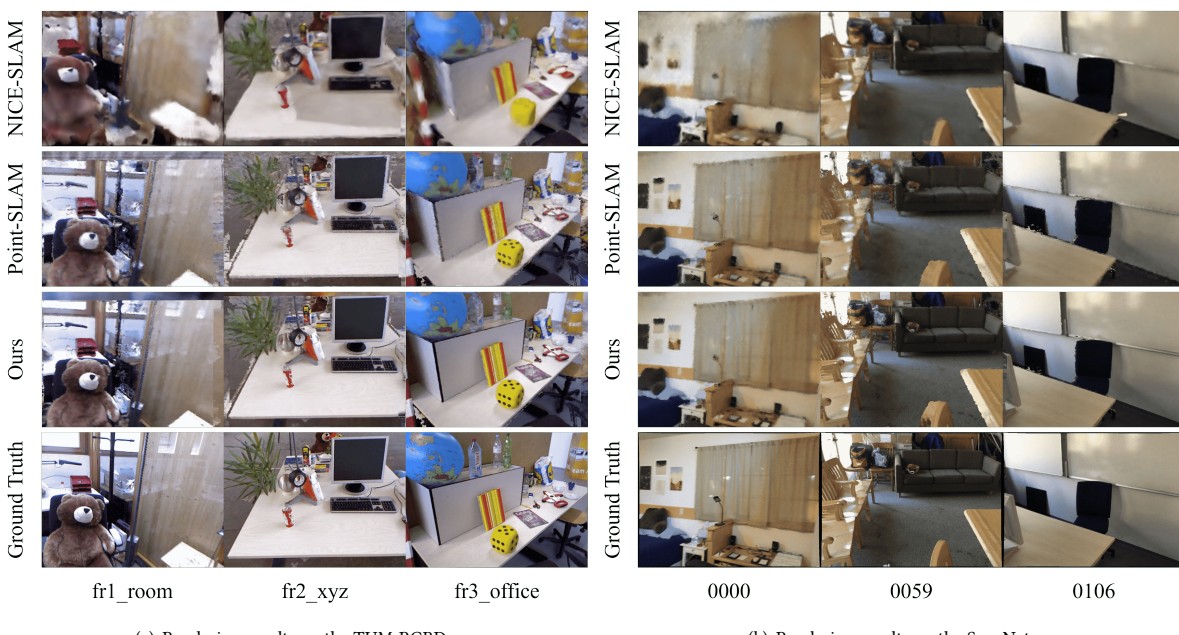

(a)  Rendering results on the TUM-RGBD

(b)  Rendering results on the ScanNet

**Figure 6: Novel view synthesis results on the real-world scenes dataset.**

| Method | 0000 | 0059 | 0106 | 0169 | 0181 | 0207 | Avg. |
|--------|------|------|------|------|------|------|------|
| DI-Fusion [13] | 62.99 | 128.00 | 18.50 | 75.80 | 87.88 | 100.19 | 78.89 |
| Vox-Fusion [44] | 68.84 | 24.18 | 8.41 | 27.28 | 23.30 | 9.41 | 26.90 |
| NICE-SLAM [51] | 12.00 | 14.00 | 7.90 | **10.90** | **13.40** | **6.20** | 10.70 |
| Point-SLAM [28] | 10.24 | 7.81 | 8.65 | 22.16 | 14.77 | 9.54 | 12.19 |
| Ours | **9.12** | **7.58** | **7.78** | 11.01 | 13.86 | 8.82 | **9.70** |

**Table 3: Tracking performance on ScanNet (ATE RMSE ↓[cm]).**

Moving to Table 2, the performance of our proposed method on the real-world TUM-RGBD dataset is presented. Outperforming all existing NeRF-based methods, our approach showcases remarkable results. For the more intricate real-world ScanNet dataset, our method achieves superior tracking results on average compared to existing NeRF-based methods as shown in Table 3. However, it's important to note that the neural point cloud exhibits sensitivity to motion blur and specularity, leading to a noticeable degradation in Point-SLAM's performance in selected sequences. In these complex scenes, our method surpasses Point-SLAM due to the incorporation

| Method | Metric | Rm 0 | Rm 1 | Rm 2 | Off 0 | Off 1 | Off 2 | Off 3 | Off 4 | Avg. |
|---|---|---|---|---|---|---|---|---|---|---|
| NICE-SLAM [51] | Depth L1 (cm) ↓ | 1.81 | 1.44 | 2.04 | 1.39 | 1.76 | 8.33 | 4.99 | 2.01 | 2.97 |
| | Precision (%) ↑ | 45.86 | 43.76 | 44.38 | 51.40 | 50.80 | 38.37 | 40.85 | 37.35 | 44.10 |
| | Recall (%) ↑ | 44.10 | 46.12 | 42.78 | 48.66 | 53.08 | 39.98 | 39.04 | 35.77 | 43.69 |
| | F1 (%) ↑ | 44.96 | 44.84 | 43.56 | 49.99 | 51.91 | 39.16 | 39.92 | 36.54 | 43.86 |
| Vox-Fusion [44] | Depth L1 (cm) | 1.09 | 1.90 | 2.21 | 2.32 | 3.40 | 4.19 | 2.96 | 1.61 | 2.46 |
| | Precision (%) ↑ | 75.83 | 35.88 | 63.10 | 48.51 | 43.50 | 54.48 | 69.11 | 55.40 | 55.73 |
| | Recall (%) ↑ | 64.89 | 33.07 | 56.62 | 44.76 | 38.44 | 47.85 | 60.61 | 46.79 | 49.13 |
| | F1 (%) ↑ | 69.93 | 34.38 | 59.67 | 46.54 | 40.81 | 50.95 | 64.56 | 50.72 | 52.20 |
| Point-SLAM [28] | Depth L1 (cm) ↓ | 0.53 | 0.22 | 0.46 | 0.30 | 0.57 | 0.49 | 0.51 | 0.46 | 0.44 |
| | Precision (%) ↑ | 91.95 | 99.04 | 97.89 | 99.00 | 99.37 | 98.05 | 96.61 | 93.98 | 96.99 |
| | Recall (%) ↑ | 82.48 | 86.43 | 84.64 | 89.06 | 84.99 | 81.44 | 81.17 | 78.51 | 83.59 |
| | F1 (%) ↑ | 86.90 | 92.31 | 90.78 | 93.77 | 91.62 | 88.98 | 88.22 | 85.55 | 89.77 |
| ESLAM [14] | Depth L1 (cm) ↓ | 0.97 | 1.07 | 1.28 | 0.86 | 1.26 | 1.71 | 1.43 | 1.06 | 1.18 |
| Ours | Depth L1 (cm) ↓ | **0.36** | **0.21** | **0.44** | **0.26** | **0.55** | **0.44** | **0.48** | **0.44** | **0.39** |
| | Precision (%) ↑ | **98.46** | **99.14** | **98.66** | **99.21** | **99.48** | **98.34** | **96.83** | **96.21** | **98.29** |
| | Recall (%) ↑ | **85.30** | **86.51** | **84.93** | **89.20** | **85.10** | **82.91** | **81.76** | **80.24** | **84.50** |
| | F1 (%) ↑ | **91.41** | **92.35** | **91.28** | **93.94** | **91.68** | **89.17** | **88.31** | **87.51** | **90.71** |

**Table 4: Reconstruction performance on Replica.**

| Method | Metric | Rm 0 | Rm 1 | Rm 2 | Off 0 | Off 1 | Off 2 | Off 3 | Off 4 | Avg. |
|---|---|---|---|---|---|---|---|---|---|---|
| NICE-SLAM [51] | PSNR (dB) ↑ | 22.12 | 22.47 | 24.52 | 29.07 | 30.34 | 19.66 | 22.23 | 24.94 | 24.42 |
| | SSIM ↑ | 0.689 | 0.757 | 0.814 | 0.874 | 0.886 | 0.797 | 0.801 | 0.856 | 0.809 |
| | LPIPS ↓ | 0.330 | 0.271 | 0.208 | 0.229 | 0.181 | 0.235 | 0.209 | 0.198 | 0.233 |
| Vox-Fusion [44] | PSNR (dB) ↑ | 22.39 | 22.36 | 23.92 | 27.79 | 29.83 | 20.33 | 23.47 | 25.21 | 24.41 |
| | SSIM ↑ | 0.683 | 0.751 | 0.798 | 0.857 | 0.876 | 0.794 | 0.803 | 0.847 | 0.801 |
| | LPIPS ↓ | 0.303 | 0.269 | 0.234 | 0.241 | 0.184 | 0.243 | 0.213 | 0.199 | 0.236 |
| Point-SLAM [28] | PSNR (dB) ↑ | 32.40 | 34.08 | 35.50 | 38.26 | 39.16 | 33.99 | 33.48 | 33.49 | 35.17 |
| | SSIM ↑ | 0.974 | 0.977 | 0.982 | 0.983 | 0.986 | 0.960 | 0.960 | 0.979 | 0.975 |
| | LPIPS ↓ | 0.113 | 0.116 | 0.111 | 0.100 | 0.118 | 0.156 | 0.132 | 0.142 | 0.124 |
| Ours | PSNR (dB) ↑ | **34.15** | **35.82** | **37.61** | **40.48** | **40.06** | **35.67** | **34.65** | **36.62** | **36.88** |
| | SSIM ↑ | **0.982** | **0.983** | **0.988** | **0.990** | **0.990** | **0.970** | **0.966** | **0.985** | **0.982** |
| | LPIPS ↓ | **0.087** | **0.095** | **0.088** | **0.068** | **0.104** | **0.126** | **0.112** | **0.107** | **0.098** |

**Table 5: Rendering performance on Replica.**

of the learnable ray transformer, which mitigates these challenges more effectively.

**Mapping Evaluation.** Table 4 presents a comparison between our method and several others, including NICE-SLAM [51], VoxFusion [44], Point-SLAM [28], and ESLAM [14], focusing on reconstruction accuracy. Across all evaluation metrics, our method outperforms all others. Specifically, we showcase an average enhancement of 87%, 84%, 11%, and 67% on the depth L1 metric in comparison to NICE-SLAM, Vox-Fusion, Point-SLAM, and ESLAM respectively. In Fig. 4, we provide a visual comparison of the mesh reconstructions generated by NICE-SLAM and our method against the ground truth. Notably, our method exhibits a significant improvement by reconstructing scene maps with higher accuracy and clarity, presenting finer details compared to previous approaches. This enhancement in reconstruction quality owes itself to two key components: the ray transformer, capable of dynamically adjusting the attention distribution to finely control the sharpness of the reconstructed surface, and the neural point cloud, which adeptly adjusts point density.

**Rendering Evaluation.** Table 5 presents a comparison of rendering results, demonstrating the superiority of our method over existing NeRF-based SLAM approaches. Additionally, in Fig. 5, Fig. 6(a), and Fig. 6(b), we showcase exemplary novel view synthesis, highlighting how our proposed SAR-SLAM generates more accurate details.

## 4.3 Memory and Runtime Analysis

We present a breakdown of the runtime and memory utilization within Table 6 for the Replica Office 0 scene. The tracking and mapping runtimes are detailed per iteration and frame. Our method exhibits a significantly smaller GPU memory footprint compared to NICE-SLAM, and slightly smaller than Point-SLAM. These runtimes were evaluated using a single Nvidia RTX 3090 Ti GPU. Furthermore, the running time and GPU/RAM memory footprint in various scenarios with different configurations are presented in the following Table 7. Despite the increase in sampled pixels, the corresponding increase in time and memory utilization remains acceptable.

| Method | Tracking /Iteration | Mapping /Iteration | Tracking /Frame | Mapping / Frame | GPU Memory Footprint |
|---|---|---|---|---|---|
| NICE-SLAM [51] | 32ms | 182ms | 1.32s | 10.92s | 11.72GB |
| Point-SLAM [28] | 21ms | 33ms | 0.85s | 9.85s | 7.98GB |
| Ours | 19ms | 29ms | 0.78s | 8.97s | 7.81GB |

**Table 6: Runtime and memory footprint of different NeRF-based SLAM methods.**

| Dataset | $M_t$ | $M$ | Tracking /Frame | Mapping / Frame | GPU Memory Footprint | RAM Memory Footprint |
|---|---|---|---|---|---|---|
| Replica | 1,500 | 5,000 | 0.78s | 8.97s | 7.81GB | 8.84GB |
| ScanNet | 5,000 | 10,000 | 1.27s | 11.91s | 9.06GB | 10.24GB |
| TUM | 5,000 | 10,000 | 1.28s | 11.32s | 8.98GB | 10.04GB |

**Table 7: Runtime and memory footprint of our method on different datasets with different settings.**

## 4.4 Ablation Study

**System Ablation.** We assess the performance of our proposed SAR-SLAM under various system configurations. Table 8 presents performance metrics obtained from the Replica Office 0 dataset. Utilizing ray transformer significantly enhances tracking, mapping, and rendering performance, underscoring the efficacy of self-attentive rendering for SLAM systems. Furthermore, employing the keyframe selection (KS) scheme and local bundle adjustment (BA) for mapping optimization leads to a further reduction in localization error. Moreover, comparing Sample Points per Ray (SPR) reveals that 5 points suffice to ensure accuracy.

**Network Ablation.** In Table 9, we present a comparison of our method across various network settings, as depicted in Fig. 3. Our proposed method (Full) outperforms the variant without the self-attention module (w/o SA), demonstrating the effectiveness of the self-attentive rendering. When changing the multi-layer architecture in the ray transformer into a single transformer layer (w/o ML), we observe a decrease in evaluation metrics to varying degrees. Additionally, excluding the Feed-Forward Network (w/o FFN) or Feature Concatenation (w/o FC) leads to less accurate rendering of depth and RGB images, consequently impacting the precision of tracking and mapping

**Keyframe Selection Ablation.** We evaluated our method using the keyframe selection strategies from iMAP and Point-SLAM, as shown in Table 10. iMAP's KS strategy randomly selects keyframes from the global scene to mitigate the scene-forgetting problem. In contrast, Point-SLAM's KS strategy maintains the keyframe list by evaluating view overlap, resulting in improved tracking accuracy and faster convergence. Unlike Point-SLAM, which utilizes sample pixels for overlap computation, our method employs the projection of neural points to obtain more effective pixels for bundle adjustment optimization, thereby enhancing tracking performance.

**Adaptive Initialization Ablation.** Fig. 7 illustrates a comparison of geometry optimization across various iteration settings for mapping initialization. Our proposed adaptive initialization method demonstrates superior performance in achieving a balance between running time and scene representation accuracy when compared to a fixed iteration number.

| Ray Transformer | KS + BA | SPR | Tracking ATE RMSE (cm) ↓ | Reconstruction Depth L1 (cm) ↓ | F1 (%) ↑ | Rendering PSNR (dB) ↑ |
|---|---|---|---|---|---|---|
| ✗ | ✗ | 5 | 1.35 | 1.26 | 78.93 | 32.06 |
| ✓ | ✗ | 5 | 0.48 | 0.29 | 93.85 | 40.34 |
| ✗ | ✓ | 5 | 0.38 | 0.30 | 93.77 | 38.26 |
| ✓ | ✓ | 3 | 0.39 | 0.28 | 93.76 | 40.16 |
| ✓ | ✓ | 5 | 0.33 | 0.26 | 93.94 | 40.48 |
| ✓ | ✓ | 7 | 0.38 | 0.28 | 93.85 | 40.19 |

**Table 8: Ablation study of our SAR-SLAM system.**

| Network Settings | Tracking ATE RMSE (cm) ↓ | Reconstruction Depth L1 (cm) ↓ | F1 (%) ↑ | Rendering PSNR (dB) ↑ |
|---|---|---|---|---|
| w/o SA | 0.51 | 0.35 | 91.89 | 37.29 |
| w/o ML | 0.47 | 0.30 | 91.66 | 38.27 |
| w/o FFN | 0.50 | 0.33 | 92.78 | 39.52 |
| w/o FC | 0.49 | 0.33 | 92.88 | 39.99 |
| Full | 0.33 | 0.26 | 93.94 | 40.48 |

**Table 9: Ablation study of network architecture.**

| KS scheme | fr1/desk | fr1/desk2 | fr1/room | fr2/xyz | fr3/office | Avg. |
|---|---|---|---|---|---|---|
| w/ iMAP | 3.10 | 7.72 | 21.33 | 6.88 | 9.32 | 9.67 |
| w/ NICE-SLAM | 2.98 | 3.14 | 18.32 | 1.88 | 3.45 | 5.95 |
| Ours | 2.79 | 3.08 | 16.86 | 1.16 | 2.89 | 5.36 |

**Table 10: Ablation study of keyframe selection scheme (ATE RMSE ↓[cm]).**

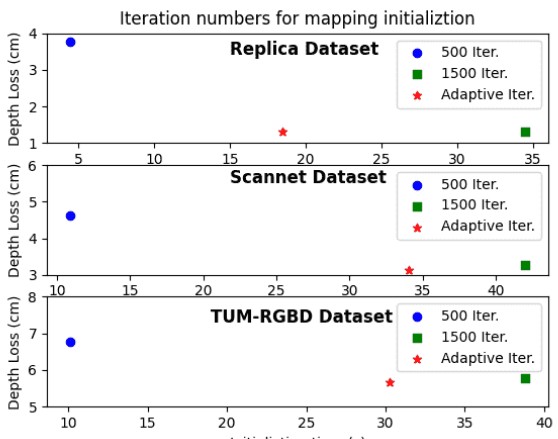

**Figure 7: Ablation study of mapping initialization.**

## 5 CONCLUSION

We introduce SAR-SLAM, a novel neural implicit SLAM system that integrates a ray transformer for view rendering and neural point cloud encoding for scene representation. The ray transformer synthesizes novel views via a self-attentive mechanism that decodes a sequence of point features sampled along the camera ray into pixel color and blending weight, enabling end-to-end learning and eliminating the necessity for fixed volume rendering equations. Experimental results on both synthetic and real-world scenes consistently demonstrate the superior performance of SAR-SLAM over existing NeRF-based SLAM methods. Notably, it exhibits enhanced tracking, reconstruction, and rendering accuracy, while also showcasing superior runtime and memory efficiency.

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

Received 20 February 2007; revised 12 March 2009; accepted 5 June 2009

