# OpenReview forum: "SAR-SLAM: Self-Attentive Rendering-based SLAM with Neural Point Cloud Encoding"
_acmmm.org/ACMMM/2024/Conference — MM2024 Poster_

### Official Review · Reviewer_f3rD · 2024-05-24

**Rating:** 4
**Confidence:** 2

**Summary:**

SAR-SLAM introduces a novel approach to simultaneous localization and mapping by integrating a ray transformer for view rendering and neural point cloud encoding for scene representation. Departing from traditional methods, it addresses limitations of existing NeRF-based SLAM approaches, offering enhanced accuracy and efficiency. Through self-attentive rendering, SAR-SLAM enables end-to-end learning, eliminating the need for fixed rendering equations. Experimental results underscore its superiority, marking a significant advancement in SLAM research.

**Strengths:**

1. The paper exhibits clear and well-organized presentation, ensuring ease of comprehension.
2. Illustrations and figures effectively elucidate the discussed concepts, aiding in the understanding of the paper's content.
3. SAR-SLAM's adoption of an implicit attention-based ray rendering approach represents a notable innovation, enhancing the system's strength, scalability, and versatility in graphical rendering.
4. Qualitative results convincingly demonstrate the effectiveness of the proposed method, highlighting its potential in SLAM applications.

**Limitations:**

1. The absence of outdoor and more challenging scene data limits the comprehensive evaluation of SAR-SLAM's performance across diverse environments.
2. A comparison with NeRF-based SLAM and 3DGS-based SLAM methodologies would provide valuable insights into SAR-SLAM's comparative advantages.
> Matsuki H, Murai R, Kelly P H J, et al. "Gaussian splatting SLAM." arXiv preprint arXiv:2312.06741, 2023.

**Suitability:**

2

---

### Official Review · Reviewer_rPHh · 2024-05-24

**Rating:** 2
**Confidence:** 4

**Summary:**

This paper propose SAR-SLAM, a point cloud encoding-based and rendering-based SLAM. Compared with previous methods, the proposed methods utilize reconstructed NeRFs and represent the scene through point cloud encoding. The proposed method achieved better performance than previsou methods on various dataset.

**Strengths:**

1) The writing is easy to follow and understand.
2) The experiments are sufficient.

**Limitations:**

1) The novelty is limited. The proposed method is similar to Point-NeRF, which is not cited in the paper. The proposed method follows previous methods, Nice-slam and point-slam, and just replace the rendering method in the pipeline with Point-NeRF. So I think the contribution is not sufficient.
2) I wonder how to get the initial point clouds? The initila point clouds are important in the rendering and localization. Are the initial point clouds robust to the noise and outliers?
3) In the implicit mapping, should the query image be close to the target image?
4) The self attention is time and memory consuming. Is there efficiency analysis?

**Suitability:**

2

---

### Official Review · Reviewer_w7Fy · 2024-05-25

**Rating:** 4
**Confidence:** 4

**Summary:**

This paper presents a simple but effective idea in the field of neural implicit SLAM: combining Point-SLAM with an IBR-style architecture that refines point features with cross attention. With the major contribution centered around the mapping part, the tracking part largely follows existing pipelines. Evaluations on common public benchmarks like Replica are presented showing improved tracking and mapping performance. Surprisingly, although this architecture adds new componenets to Point-SLAM, the efficiency is improved in terms both speed and storage. This involves good engineering efforts and I am looking forward to the code.

**Strengths:**

+ The idea is simple but very effective as shown by the experiments. Interpolation in point based representation may be suboptimal and using a data-driven attention-based module to mix features from the neighborhood makes sense. This leads to a combination of Point-SLAM and IBR but this design differs from IBR in nature. IBR mixes features from views while this one mixes features from the neighbors. This is considered as a novel usage of an existing module in another (reasonable) setting.
+ The experiments are quite comprehensive, showing good ablations and benchmark results. A notable thing is that while this design incorpoarates new modules into Point-SLAM, efficiency in terms of both speed and storage are improved. The field of SLAM naturally favors this kind of engineering efforts.

**Limitations:**

- The effectiveness of this method seems to originate from the principle idea: using IBR attention to improve the feature mixing in a Point-NeRF. I think this is not specific to the SLAM setting and also applies to the original Point-NeRF setting. Could you please provide another ablation study that investigates whether this architecture works for generic novel view synthesis quality of Point-NeRF or Point-NeRF variants? I think this is important to reach the conclusion how this paper brings knowlege to the whole neural rendering community.
- A notable advantage of this paper its efficiency caused by good engineering. But for now, the (anonymous) code is not released yet, which is a pity. This is an important factor for SLAM papers.
- I recommend two references on neural rendering effiency [A] and multis-cale rendering quality [B], which is related to the paper. Could you please kindly consider include and discuss them in the paper?

[A] SlimmeRF: Slimmable Radiance Fields, 3DV 2024

[B] Rip-NeRF: Anti-aliasing Radiance Fields with Ripmap-Encoded Platonic Solids, SIGGRAPH 2024

**Suitability:**

3

---

### Meta-Review · Area_Chair_Cmam · 2024-07-04

**Recommendation:** Accept (Poster)
**Confidence:** 4

**Metareview:**

The paper introduces a new approach to simultaneous localization and mapping which addresses some limitations (reliance on volume rendering techniques, which oversimplify the modelling) of existing NeRF-based SLAM approaches, leading to improved accuracy and efficiency.

The rebuttal addressed the main concerns raised by reviewers.